# The final walk with preptin

**Lucie Mrázková[1,2], Marta Lubos[1], Jan Voldřich[1], Erika Kužmová[1], Denisa Zrubecká[1], Petra Gwozdiaková[1], Miloš Buděšínský[1], Seiya Asai[1], Aleš Marek[1], Jan Pícha[1], Michaela Tencerová[3], Michaela Ferenčáková[3], Glenda Alquicer Barrera[3], Jakub Kaminský[1], Jiří Jiráček[1], Lenka Žáková[1]***

1 Institute of Organic Chemistry and Biochemistry of the Czech Academy of Sciences, Prague, Czech Republic, 2 Faculty of Science, Department of Cell Biology, Charles University, Prague, Czech Republic, 3 Institute of Physiology of the Czech Academy of Sciences, Prague, Czech Republic

* lenka.zakova@uochb.cas.cz

**Data Availability Statement:** All relevant data are within the manuscript, its Supporting Information files or are available via the OSF data repository (https://doi.org/10.17605/OSF.IO/JS8RA).

## Abstract

Preptin, a 34-amino acid peptide derived from pro-IGF2, is believed to influence various physiological processes, including insulin secretion and the regulation of bone metabolism. Despite its recognized involvement, the precise physiological role of preptin remains enigmatic. To address this knowledge gap, we synthesized 16 analogs of preptin, spanning a spectrum from full-length forms to fragments, and conducted comprehensive comparative activity evaluations alongside native human, mouse and rat preptin. Our study aimed to elucidate the physiological role of preptin. Contrary to previous indications of broad biological activity, our thorough analyses across diverse cell types revealed no significant biological activity associated with preptin or its analogs. This suggests that the associations of preptin with various diseases or tissue-specific abundance fluctuations may be influenced by factors beyond preptin itself, such as higher levels of IGF2 or IGF2 proforms present in tissues. In conclusion, our findings challenge the conventional notion of preptin as an isolated biologically active molecule and underscore the complexity of its interactions within biological systems. Rather than acting independently, the observed effects of preptin may arise from experimental conditions, elevated preptin concentrations, or interactions with related molecules such as IGF2.

## Introduction

In the late 1960s, two independent groups [1,2] postulated the theory that a mature peptide is formed from a longer polypeptide after proteolytic cleavage at specific sites. Furthermore, it is possible that several different peptides with different biological activities can be formed from one initial polypeptide [3,4]. The precursor of IGF2, pro-IGF2, is 156 amino acids long and takes 67 amino acids to convert to mature IGF2. It must be properly glycosylated and cleaved by specific proteases as a part of post-translational modifications [5,6]. The processing of the remaining 89-amino acid E-domain forms a 34-amino acid peptide called preptin, which corresponds to amino acids 69–104 of pro-IGF2.

**Funding:** The work was supported by the Czech Science Foundation (grant No. 19-14069S (to LZ) and 22-12243S (to MT)), by the National Institute for Research of Metabolic and Cardiovascular Diseases (Program EXCELES, ID Project No. LX22NPO5104) - Funded by the European Union – Next Generation EU and by the Academy of Sciences of the Czech Republic (Research Project RVO:6138963, support to the Institute of Organic Chemistry and Biochemistry). the funders had no role in study design, data collection and analysis, decision to publish, or preparation of the manuscript.

**Competing interests:** The authors have declared that no competing interests exist.

Preptin was first discovered in pancreatic beta cells [7]. Previous research showed that preptin can influence insulin secretion, thus carbohydrate metabolism, and also acts as a bone anabolic agent [7–11]. Furthermore, a correlation between preptin levels and metabolic or bone diseases in humans has been shown. Preptin levels positively correlate with diabetes type II, impaired glucose tolerance, polycystic ovary syndrome, obesity, and gestational diabetes [12–18]. On the contrary, preptin levels were found to be low in patients with low bone mineral density and osteoporosis. [19,20]. Moreover, while some *in vitro* studies on bone cells have shown a positive effect of preptin and its analogs or fragments on osteoblast proliferation and activity [8,9,11], some other studies have been more ambiguous, even using similar cellular systems [10,17].

In this study, we tried to shed more light on all the biological effects of preptin and its fragments described so far. Since it has been shown that not only preptin but also preptin fragments exhibit some biological potency, we synthesized 16 preptin analogs and compared their activities to those of native human, mouse and rat preptin. We aimed to synthesize a series of analogs that would cover i) full-length preptin with various modifications, then ii) C-terminal, N-terminal and middle fragments of preptin, and iii) also analogs with a stabilizing cycle that could give the analog a more secondary structure (Fig 1). In preparing these cyclic analogs, we followed the methodology described in our previous publication [21]. We tested all derivatives at all receptors involved in the insulin protein family and especially the IGF2 receptor, which has been earlier suggested as a putative receptor for preptin [22]. In addition, to reveal the receptor for preptin, we performed several binding experiments with radiolabeled preptin on different cells that are relevant for preptin activity. All our results allowed us to better clarify the role of preptin in the organism.

## Materials and methods

### Synthesis of preptins and analogs

The full-length preptins and their analogs (Fig 1) were synthesized on the Spyder Mark IV multiple Peptide Synthesizer (European Patent application EP17206537.7), developed in the Development Center of the Institute of Organic Chemistry and Biochemistry (Development Center of the Institute of Organic Chemistry and Biochemistry of the CAS (DCIOCB), http://dc.uochb.cz) on Rink Amide AM resin or preloaded Wang resins, using the Fmoc/tBu protecting strategy and HBTU/DIPEA and DIC/HOBT activation.

Unnatural amino acids Fmoc-L-Lys(N₃)-OH [23], Fmoc-L-Pra-OH and olefinic Fmoc-protected amino acids–Fmoc-5S-OH, Fmoc-5R-OH, Fmoc-L-Cys(allyl)-OH, Fmoc-D-Cys(allyl)-OH were attached manually during the synthesis on the resin. Synthesis of Fmoc-L-Cys(allyl)-OH and Fmoc-D-Cys(allyl)-OH is shown in SI in Schemes S1 and S2 and S1-S3 Figs in S1 Data. Due to steric hindrance, amino acids following olefinic residues were also attached manually. To minimize aspartimide by-products, we introduced Fmoc-Asp (OEpe)-OH derivative (attached manually) in Asp-Asn-containing peptides; synthesis of Fmoc-Asp (OEpe)-OH is shown (S4 Fig in S1 Data).

The peptide syntheses were performed according to the protocols previously described in Lubos et al. [21]. The purity of each peptide was checked by reverse-phase high-performance liquid chromatography (RP-HPLC) on the Watrex HPLC system (Watrex DeltaChrom™ P200 binary Pump and Wufeng LC-100 UV Detector), using a Nucleosil 120–5 C8 column (250 × 4.6 mm, 5 μm, Macherey-Nagel) at a flow rate of 1 mL/min. The following solvent system was used: Solvent A: 0.1 % TFA (v/v) in H₂O; Solvent B: 0.1 % TFA (v/v) in 80% CH₃CN. The following gradient was used: t = 0 min/10 % B, t = 30 min/100 % B. The compounds were detected at 218 nm and the purity of the compounds was checked at this wavelength.

| Peptide | | Sequence |
|---|---|---|
| **1** | human preptin | DVSTPPTVLPDNFPRYPVGKFFQYDTWKQSTQRL |
| **2** | | DVSTPPTVLPDNFPRY |
| **3** | | PVGKFFQYDTWKQSTQRL |
| **4** | | DVSTPPTVLPDNFPRY-amide |
| **5** | | DVSTPPTVLPDNFPRYPVGK*F*FQYDTWKQSTQRL |
| **6** | | DVSTPPTVLPDNFPRX(16)PVGX(20)FFQYDTWKQSTQRL-amide  |
| **7** | | DVSTPPTVLPDNFPR*X*(16)PVGX(20)FFQYDTWKQSTQRL-amide  |
| **8** | | DVSTPPTVLPDN*X*(13)PRYPVGX(20)FFQYDTWKQSTQRL-amide  |
| **9** | | TW*X*(3)QST*X*(6)RL  |
| **10** | | TW*X*(3)QST*X*(6)RL  |
| **11** | | TW*X*(3)QSTX(6)RL  |
| **12** | | WKQSTQRL-amide |
| **13** | | TWKQSTQRL-amide |
| **14** | mouse preptin | DVSTSQAVLPDDFPRYPVGKFFQYDTWRQSAGRL |
| **15** | | DVSTSQAVLPDDFPRYPVGZFFQYDTWRQSAGRL |
| **16** | | DVSTSQAVLPDDFPRY |
| **17** | | DVTTSQAVLPDDFPRY |
| **18** | | FPRYPVGKFFQYNTW-amide |
| **19** | rat preptin | DVSTSQAVLPDDFPRYPVGKFFKFDTWRQSAGRL |

**Fig 1. Preptins and preptin derivatives synthesized.** Sequences are shown in single-letter codes. C-terminal carboxamides are depicted as -amide. The position with non-standard amino acids, precursors for ring-closing olefin metathesis (RCM) or Cu(I)-catalyzed azide-alkyne cycloaddition (CuAAC) reactions are shown as X and numbered, and the respective cyclized segments also shown as ChemDraw structures. (R)-amino acids are shown in standard single-letter codes (or as X) but in italics. α-Amino isobutyric acid in **15** is shown as Z.

Preparative RP-HPLC chromatography was carried out on the Waters HPLC system (Waters 600 with 2487 Dual λ Absorbance Detector) using a Nucleosil 100–7 C8 column (250 × 10 mm, 7 μm, Macherey-Nagel) at a flow rate of 4 mL/min, with the same gradient and solvent system as mentioned above.

The analytical data (mass spectra and HPLC traces) for preptin derivatives **1**–**19** are shown (S5-S42 Figs in S1 Data). The NMR assessment of configuration of stereochemistry of compounds **10** and **11** with dicarba bridges is shown (S1 and S2 Tables in S1 Data). S3-S5 Tables in S1 Data show results of analyses of peptides **1**–**19** by circular dichroism.

## Ring-closing olefin metathesis (RCM)

Ring-closing olefin metathesis was performed according to the protocol previously reported in Lubos et al. [21]. For peptide **7**, Grubbs' first-generation catalyst in DCM was used, while for peptides **10** and **11**, Hoveyda-Grubbs' second-generation catalyst in DCE was used.

## Cu(I)-catalyzed azide-alkyne cycloaddition (CuAAC, click reaction)

Click reaction was performed according to the protocol previously reported in Lubos et al. [21].

## Circular dichroism and secondary structure assessment

Circular dichroism (CD) spectra of all peptides **1**–**19** investigated (Fig 1) were recorded using a JASCO J-818 spectrometer. The peptides were dissolved in phosphate buffer (pH 7.2) to 0.1 mg/ml final concentration. The spectra were recorded in the wavelength range of 190–250 nm and the resulting spectra represent the average of three scans. All experiments were performed using a 1 mm quartz cell (Hellma Analytics) at room temperature, scan rate 5 nm/min and response time 16 s. Solvent signal was subtracted from final spectra. CD spectra were normalized to the concentration, the length of the cell and the number of amino acids in the peptide. (S3 Table in S1 Data) summarizes the details of each sample. Recorded spectra were further analyzed using the BeStSel program [24,25] to determine the secondary structure of peptides **1**–**19**. The program decomposes the analyzed spectrum into a set of basic spectra corresponding to proteins with known structure. Using a linear combination of these basic spectra, the secondary structure of the peptide under study is determined. BeStSel considers eight different secondary structure motifs: two different a-helices, three antiparallels and one parallel β-strand, a turn motif and a conformations labeled "other".

## Cell cultures

Mouse embryonic fibroblasts used for binding and signaling assays were derived from animals with targeted disruption of the IGF1 receptor gene [26] and stably transfected with expression vectors containing either A (R−/IR-A) isoforms of human insulin receptor or human IGF-1 receptor (R+39) [27,28]. As a model for the determination of binding affinities of peptides towards M6P/IGF2R, we used non-transfected R- cells that contain predominantly only this receptor [5]. The R−/IR-A, R+39 and R-cell lines were kindly provided by A. Belfiore (University of Magna Graecia, Catanzaro, Italy) and R. Baserga (Thomas Jefferson University, Philadelphia, PA). Cells were grown in DMEM medium with 5 mM glucose (Biosera), supplemented with 10 % fetal bovine serum, 2 mM L-glutamine, 0.3 μg/mL puromycin (not added to R- cells), 100 units/mL penicillin and 100 μg/mL streptomycin in humidified air with 5 % $CO_2$ at 37˚C.

Mice preosteoblast MC3T3-E1 cells (subclone 4) were purchased from ATCC®. Cells were grown in MEM medium, supplemented with 10 % fetal bovine serum, 2 mM L-glutamine, 100 units/mL penicillin and 100 µg/mL streptomycin in humidified air with 5 % $CO_2$ at 37˚C. Human osteosarcoma U-2 OS cells were purchased from ATCC®. Mice pancreatic beta cells line MIN6 were purchased from Japan (kind gift from Miyazaki Laboratory, Osaka University, Japan). U-2 OS cells were grown in DMEM medium, supplemented with 10 % fetal bovine serum, 2 mM L-glutamine, 100 units/mL penicillin, and 100 µg/mL streptomycin in humidified air with 5 % $CO_2$ at 37˚C. MIN6 cells were cultured in the same complete medium as U-2 OS, but they were supplemented in addition with 2-mercaptoethanol at the final concentration 50 µM. Human embryonic kidney HEK-293 cells with stable transfection with F22 plasmid, in our paper are called cAMP-HEK cells (GloSenzor, Promega, USA) (cells were a kind gift from David Sedlák, Open Screen, IMG CAS). Cells were grown in DMEM medium, supplemented with 10 % fetal bovine serum and 2 mM L-glutamine in humidified air with 5 % $CO_2$ at 37˚C.

A telomerized MSC line (hBMSC-TERT) was used as a model for bone marrow-derived MSCs, as previously described [29]. hBMSC-TERT were maintained in Minimum Essential Medium (MEM), supplemented with 10 % fetal bovine serum (FBS) and 100 units/mL penicillin, and 100 µg/mL streptomycin (Gibco-Invitrogen, U.S.A).

## Saturation binding experiments of [$^{125}$I]-monoiodotyrosyl-preptin (mouse) on various types of cells

To get an initial insight into what cells preptin can bind to, we used a [$^{125}$I]-monoiodotyrosyl-variant of mouse preptin **14** prepared in our laboratory (the preparation described in SI). For this, we used cell lines that are representative of tissues that are predicted to be affected by preptin. We used E1 preosteoblasts, U-2 OS osteoblasts, MIN6 pancreatic β-cells, and non-transfected R-cells containing the IGF2R.

For this, 14 000 of the respective cells were seeded in each well in complete medium 24 hours before the experiment. The cells were incubated in a total volume of 250 µl of a binding buffer (100 mM HEPES pH 7.6, 100 mM NaCl, 5 mM KCl, 1 mM EDTA, 1.3 mM $MgSO_4$, 10 mM glucose, 15 mM sodium acetate and 1 % bovine serum albumin) with various concentrations (0–7 nM) of [$^{125}$I]-monoiodotyrosyl-preptin (2,200 Ci/mmol), for 16 h at 5˚C. Thereafter, the wells were washed twice with saline buffer. The bound proteins in the wells were solubilized twice with 500 µl of 0.1 M NaOH that was collected. Bound radioactivity was determined in the γ-counter. Nonspecific binding was determined by measuring the remaining bound radiotracer in the presence of 10 µM unlabeled preptin for each tracer concentration. Each experiment was performed in monoplicates two or three times and the results were evaluated in GraphPad Prism 8, using non-linear regression considering binding to one site.

## Determination of binding affinities for IGF1R, IR-A and IGF2R

We also tested whether non-labeled preptins can bind to any of receptors for IGF2 (i.e. IR, IGF1R and IGF2R) in competitive binding assays, employing $^{125}$I-labeled human insulin, IGF1 or IGF2. To determine binding to the IGF1R, we used mouse fibroblasts transfected with human IGF1R and with deleted mouse IGF1R (R+39 cells), according to Hexnerova et al. [30]. We used non-transfected 3T3 fibroblasts to determine binding to IGF2R, because these cells contain essentially only IGF2R [5] and we used human IM-9 lymphocytes to determine affinity for IR-A [31].

Binding affinities of ligands for IGF1R were determined with [$^{125}$I]-Iodotyrosyl-IGF1 (2,614 Ci/mmol) prepared in the IOCB radioisotope laboratory by a procedure described in Kertisova et al. [32]. Binding affinities for IR-A were determined with [$^{125}$I]-monoiodotyrosyl-

TyrA14-insulin (2,200 Ci/mmol), prepared as described by Asai et al. [33]. And [$^{125}$I]-monoio-dotyrosyl-Tyr2-IGF2 (2 200 Ci/mmol) [34] was used as a radiotracer to determine the binding affinity of ligands to IGF2R.

## Analysis of binding data and statistical evaluation of binding affinities

The dissociation constant ($K_d$) values were determined with GraphPad Prism 8, using a non-linear regression method, a one-site fitting program and considering the potential depletion of free ligand. The individual binding curves of each peptide for each receptor were determined in duplicate points, and the final dissociation constant ($K_d$) was calculated from at least three (n≥3) binding curves (each curve giving a single $K_d$ value), determined independently and compared to binding curves for human insulin or human IGF-1, depending on the type of receptor. Relative binding affinities were calculated as ($K_d$ of the native hormone/$K_d$ of analog) × 100 (in %). The relative binding affinity S.D. values of analogs were calculated as S.D. = $K_d$ of the native hormone/$K_d$ of peptide ×100 × $\sqrt{[(\text{S.D. native}/K_d \text{ native})^2 + (\text{S.D. analog}/K_d \text{ analog})^2]}$.

## Signaling pathways activation

For measuring activation of intracellular signaling pathways, 40 000 of MC3T3-E1, U-2 OS and 3T3-R- cells per well and 100 000 of MIN6 cells per well were seeded in a 24-well plate, two days before stimulation. Cells' stimulations were done after overnight starvation in a clean culture medium without serum. Cells were stimulated using preptins and their analogs at $10^{-8}$M concentration for 5 min at RT. Cells were washed with PBS and froze immediately. Samples for immunoblots were prepared in Sample lysis buffer (62,5 mM Tris/HCl, 2 % SDS (w/v), 10 % glycerol (v/v), 0,01 % Brom-phenol Blue (w/v), 0,1M DTT, 50mM NaF, 1mM Na$_3$VO$_4$, 0,5 % protease inhibitory complex (v/v), pH 6,8).

Western blots were similarly used to study signaling pathways activated by our preptins in all cell lines mentioned above. Proteins were routinely analyzed using immunoblotting. The PVDF membranes were probed with anti-phospho-Akt (Thr308) (C31E5E) Rabbit mAb (Cell Signaling Technology #2965, dilution 1:1000), anti-phospho-p44/42 MAPK (Erk 1/2) (Thr202/Tyr204) (E10) Mouse mAb (Cell Signaling Technology #9106, dilution 1:2000), anti-PI3K p110α (C73F8) antibodies (Cell Signaling Technology #4249, dilution 1:1000) and anti-GAPDH antibodies as a loading control (Cell Signaling Technology #97166, dilution 1:1000). Each experiment was repeated 4 times. The data are expressed as the contribution of target phosphorylation relative to non-stimulated cells (MC3T3-E1, U-2 OS, 3T3-R- and MIN6). Mean ± S.D. (n = 4) values were calculated. The significance of the changes in stimulation of phosphorylation in relation to unstimulated cells was calculated, using the One-way-ANOVA comparing all preptins and their analogs versus control non-stimulated cells.

## Calcium release

The day before the experiment, U-2 OS, MC3T3-E1, MIN6 and cAMP-HEK cells were seeded 6 000 (for osteoblastic cell lines U-2 OS, MC3T3-E1), 10 000 (for cAMP-HEK cells) or 30 000 (MIN6) cells per well in a black 384-well format micro plate with transparent bottom (Cat.#: 142761, Nunc, Thermo Fisher Scientific, USA). On the next day, the cultivation media was exchanged for assay buffer, consisting of HBSS (Cat.#: H6648-500ML, Sigma-Aldrich, Merck, Germany), 20 mM HEPES (Cat.#: L0180-100, Biowest, France), 2.5 mM Probenecid (Cat.#: P8761, Sigma-Aldrich, Merck, Germany), adjusted for pH 7.4 and supplemented with 4.77 μM CalciFluor™ Fluo-8, AM (Cat.#: 1345980-40-6, Santa Cruz Biotechnology, USA). After 45 min incubation with assay buffer in 37˚C and 5% CO$_2$, the cells were incubated at RT in a dark

place for another 15 minutes. After this time, preptin and its analogs were added by the Agilent Bravo Automated Liquid Handling Platform (Agilent Technologies, USA). The kinetics were measured by the Spark® multimode reader from Tecan (Tecan Group Ltd., Switzerland) for 10 min (approximately 45 cycles, excitation 485 nm, emission 535 nm, band width 10 nm). Before each kinetics, the background fluorescence ($F_0$) was measured and the result was calculated as $(F_{max}-F_0)/F_0$, where $F_{max}$ corresponds to the maximum response upon stimulation by the compounds. Each compound was tested in concentration $10^{-4}$ M.

## cAMP measurement

Intracellular cAMP concentration was measured using Promega's Glosensor on osteoblasts U-2 OS, pancreatic β-cells MIN6 and control cAMP-HEK cells (with stable transfection of F22 plasmid as described above). U-2 OS cells were transfected with F22 plasmid (pGloSensor™-22F, Promega, USA, 6.1 μg of plasmid per 1 million cells) using X-tremeGENE (Sigma-Aldrich, Germany). The ratio of plasmid and transfection reagent was 1:3 (plasmid: X-tremeGENE, m/v). The MIN6 cells were transfected with F22 plasmid in the same concentration as U-2 OS using electroporation by the Invitrogen™ Neon™ Transfection System Instrument (Thermo Fisher Scientific, USA, transfection conditions: 1400 V, 20 ms, 1 pulse). The cells were then seeded at 6 000 (U-2 OS), 60 000 (MIN6) and 10 000 (cAMP-HEK cells) cells per well in a white 384-well plate with white bottom (Cat. #: 164610, Nunc, Thermo Fisher Scientific, USA) and incubated for 24 (U-2 OS, cAMP-HEK cells) or 48 hours (MIN6 cells). For seeding of cAMP-HEK cells, coating of wells with poly-D-lysin was necessary.

After 24 hours (U-2 OS and cAMP-HEK cells) or 48 hours, respectively (MIN6 cells), cultivation media was changed for assay buffer (HBSS buffer with 20 mM HEPES and 2% (v/v) (cAMP-HEK, U-2 OS cells) or 6% (v/v), respectively (MIN6 cells), GloSensor™ reagent stock solution). The plate was incubated in the dark for 1–2 hours at RT and luminescence was measured until a stable signal was observed ($L_0$). After this time, preptin and its analogs were added by the Agilent Bravo Automated Liquid Handling Platform. The luminescence kinetics were measured by Spark® for 30 min. The result was calculated as $(L_{max}-L_0)/L_0$, where $L_{max}$ corresponds to the maximum response upon stimulation by the compounds. Each compound was tested in 5 concentrations ($10^{-8}$–$10^{-4}$ M). Stock solutions were pre-spotted in a separate 384-well plate using Echo 650 (Beckmann Coulter, USA), diluted and added directly to the cells cultivated in the 384-well plate, using the Agilent Bravo Automated Liquid Handling Platform (Agilent Technologies, USA).

## Stimulation of insulin secretion

MIN6 cells were seeded at a density of 100 000 cells/well in 24-well culture dishes and grown for 3 days in 1 ml of complete medium [33]. Before experiments, cells were maintained for 2 h in a glucose-free medium. Thereafter, the cells were washed twice and preincubated for 2 h at 37˚C in 0.4 ml glucose-free Krebs-Ringer bicarbonate HEPES (KRBH) buffer of the following composition: 120 mM NaCl, 4.6 mM KCl, 2 mM $CaCl_2$, 5 mM $NaHCO_3$, 1 mM $MgSO_4$, 0.4 mM $KH_2PO_4$, 0.15 mM $Na_2HPO_4$, BSA (0.2 % w/v), and 20 mM HEPES with pH value adjusted to 7.4 with NaOH. Next, the buffer was removed from cells and a new portion of 0.4 ml of glucose-free KRBH buffer, but with a respective concentration of glucose and/or preptin, was added to the cells, which were incubated for 1 h at 37˚C. The cells were stimulated with various concentrations of D-glucose (0 and 10 mM) and the effects of various concentrations of mouse preptin were determined for 10 mM D-glucose concentration. After the stimulation, supernatants were collected from the cells and stored at −80˚C before the measurement of insulin concentration.

## Radioimmunoassay

Radioimmunoassay (RIA) kits for the measurement of rat insulin (sensitive Rat Insulin RIA Kit, Cat. # SRI-13K) content in samples were provided by EMD Millipore Corporation and were used according to the manufacturer's instructions. Each biological sample was measured in duplicate.

## Osteoblast (OB) differentiation

hBMSC-TERT were plated at a density of 20 000 cells/cm$^2$ (Alizarin Red S staining) and 5 000 cells/well (ALP activity assay) in MEM medium (Gibco), supplemented with 10 % FBS (Gibco) and 1 % P/S (Gibco). One day after seeding, the medium was replaced with OB induction medium, composed of basal medium, supplemented with 10 mM B-glycerophosphate (Sigma-Aldrich), 10 nM dexamethasone (Sigma-Aldrich), 50 μg/mL vitamin C (Sigma-Aldrich), 10 nM vitamin D3 (Sigma-Aldrich) and treated with 100 ng/ml, 500 ng/ml and 1 ug/ml IGF2, IGF1, IGF2, preptin or vehicle (acetic acid) as a control. The medium was changed every other day for 7 days (ALP activity assay) or 10 days (Alizarin Red S staining).

## Alizarin Red S staining

Mineralized matrix formation at Day 10 of OB differentiation was measured, using Alizarin Red S staining [35]. Cells were fixed with 70 % ice-cold ethanol for 1 h at -20˚C, before addition of Alizarin Red S Solution (AR-S) (Sigma-Aldrich). The cells were stained for 10 min at room temperature (RT). Excess dye was washed with distilled water and with phosphate buffered saline (PBS) (Gibco) to reduce nonspecific AR-S stain.

## Alkaline phosphatase (ALP) activity assay

ALP activity and cell viability assay were quantified at Day 7 of OB differentiation, in order to normalize the ALP activity data to the number of viable cells.

Cell viability assay was measured using a Cell Titer-Blue Assay Reagent (Promega) at fluorescence intensity ($579_{Ex}/584_{Em}$). ALP activity was detected by absorbance at 405 nm, using p-nitrophenyl phosphate (Sigma-Aldrich) as substrate [35].

## Ethics statement

This article does not contain any studies with human or animal subjects performed by any of the authors.

# Results and discussion

## Peptide synthesis and covalent peptide modification

Human preptin and its analogs were synthesized using solid phase peptide synthesis and products were purified and analyzed on HPLC and their identities confirmed using MS (S5-S42 Figs in S1 Data). It was not easy to couple some non-biogenic amino acids, but after overnight reaction and analytical microcleavage followed by LC-MS analysis, we could confirm that couplings were successful. Covalent modifications (cyclizations with RCM and CuAAC click reaction) usually proceeded without problems.

In this study, we prepared 19 peptides: native human (peptide **1**), mouse (peptide **14**), rat preptin (peptide **19**), 12 analogs or fragments of human preptin (peptides **2–18**) in Figs 1 and 4 analogs or fragments of mouse preptin (peptides **15–18**) to test hypotheses about their action in different tissues.

## Secondary structures of peptides

The secondary structure of the prepared peptides **1**–**19** was investigated by circular dichroism. Fig 2 shows the spectra of all peptides divided into three groups. The first (Fig 2A) consists of human preptin **1** and its six analogs or fragments (peptides **2**–**5**, **12**–**13**). Stapled peptides derived from human preptin (peptides **6**–**11**) make up the second group (Fig 2B). Finally, the third group (Fig 2C) consists of mouse preptin **14** and its analogs (peptides **15**–**18**) and rat preptin **19**. In (S3 Table in S1 Data) we summarizes the detailed characteristics (wavelength minima and their intensities) of the CD bands observed for peptides **1**–**19**. Note that all spectra have been normalized to concentration, path length, and number of amino acids. All spectra in Group 1 have a relatively uniform shape, having a single broad negative band with a minimum at ~199 ± 1 nm. The intensity of this band ranges from De -2.0 for peptide **3** to -4.2 for peptide **4**. Interestingly, there is no apparent correlation between peptide length and CD signal intensity. The intensity for peptide **5**, which has 34 amino acids, is even lower than the intensity for peptides **12** and **13**, which have only 8 and 9 amino acids. In conclusion, increased chain length does not necessarily provide a significant enhancement in CD in these peptides. The observed CD shape for peptides in Group 1 is characteristic of disordered peptides. This is confirmed by the secondary structure estimates based on the BeStSel analysis. S5 Table in S1 Data summarizes the estimated content of eight different structural motifs for peptides **1**–**19**. Disordered structures are classified as "other" within the BeStSel model. We see a relatively high proportion of disordered structures for peptides from Group 1. The small content of α-helices ($< 8\%$) seen for some of the longer peptides in the group (**1**, **2**, **4**) is completely reduced for the very short peptides **12** and **13**. The observed CD shapes and the secondary structure estimates are consistent with previously reported CD spectra of preptin fragments [21].

The spectra of stapled peptides **6**–**11** differ more, as reported earlier for short (8 or 9 amino acids) stapled preptin fragments [21]. Peptides **10** and **11** are structurally very close to peptide **5** from Ref. [21], but a difference in the absolute configuration on the amino acid of the staple and the nature of the staple can be noticed. In this work, peptides **6** and **8** provide CD similar to spectra of the aforementioned Group 1 with one broad negative band. Peptides **9** and **11** have the minimum of this band slightly blue–shifted by ~5 nm. The intensity of this band is also lower. In addition, peptides **9** and **11** show a weak positive maximum at ~223 nm. The CD band minimum of peptide **10** is even more shifted down to 191 nm and the intensity of the positive band is twice as high as that of peptide **9** or **11**. The shift of the band to lower wavelengths and the new positive band at ~223 nm may be an indication of increased content of polyproline II structure (PPII), and was discussed in Lubos et al. [21] for short preptin fragments. It has been reported that disordered peptides have a strong structural bias for helical PPII [36,37]. Note that the presence of a higher amount of proline in the peptide is not necessary for PPII [38]. The different CD shape of peptides **9**–**11** compared to peptides **6** and **8** may

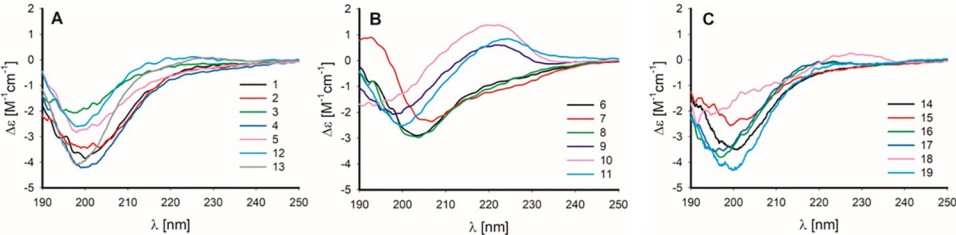

**Fig 2. CD spectra of peptides 1–19 recorded in phosphate buffer.** Peptides were divided into three groups: Human preptin analogs (**A**), stapled human preptin analogs (**B**), mouse and rat preptin analogs (**C**).

be due to the higher abundance of a different type of b-strand conformation (right-handed, see (S5 Table in S1 Data)). In contrast to the other peptides in the group, peptide **7** shows a +/-/- CD pattern, typical of helical peptides. However, its intensity is relatively low and therefore it is unlikely to be a highly helical structure. And indeed, the estimated helical content, although the highest of all peptides, is only 14%, as shown (S5 Table in S1 Data).

Mouse (**14**) and rat preptin (**19**) and analogs and fragments of mouse preptin (**15–18**) give CD spectra similar to peptides derived from human preptin (**1–5**). This would suggest that they are also highly disordered, although some PPII content cannot be ruled out, especially for **16** and **17**.

## Neither saturation nor competition experiments show potent binding of preptin to cells

Firstly, we prepared $^{125}$I-human preptin and we used this tracer for saturation experiments (preparation of $^{125}$I-human preptin is in SI). We performed saturation binding experiments to directly see if labeled preptin could bind to cells on which it would have a putative biological effect, i.e. on pancreatic, bone and IGF2 receptor-containing cells. Saturation experiments showed us a very low ability of preptin to bind to any type of cells we used. Bound radioactivity even at concentrations as high as 7 nM of the radioligand showed low values (S43 Fig in S1 Data). Moreover, levels of non-specific binding were more than 50% of the total bound $^{125}$I-preptin for all cell types. Therefore, the specific bound activity could not be relevantly determined. From competitive studies, we wanted to determine whether cold preptin could be at least a weak competitor of natural ligands (insulin, IGF1 and IGF2) for the receptors for insulin, IGF1 and IGF2 (Fig 3). Whereas on R- with IGF2R and fibroblasts with IGF1R, preptin did not displace native ligands even at the highest concentrations used; on IR-A in IM-9 lymphocytes, human preptin (peptide **1**) and the N-terminal fragment of preptin (peptide **3**) showed weak binding, but only at the concentrations > $10^{-6}$M.

Thus, overall, it is unlikely that preptin could trigger its biological activity via IGF2R, as suggested by Cheng et al. [22]. Neither IGF1R nor IR-A appear to be the target receptors for preptin. Unfortunately, the results from the saturation curves on bone and pancreatic cells do not seem to indicate that these cells contain any other high-affinity target receptor for preptin. It is possible, however, that preptin acts through a receptor that is less abundant on the surface of these cells or a low-affinity receptor that is, despite weak binding, able to cause some biological effect.

## Stapled preptins show significantly lower signaling ability on bone cells

Since we did not observe any relevant binding in the binding assays, we wanted to test whether preptins could activate any of the signaling pathways. It has been reported that preptin-

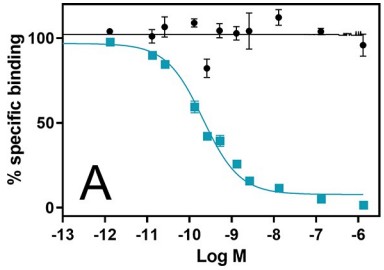 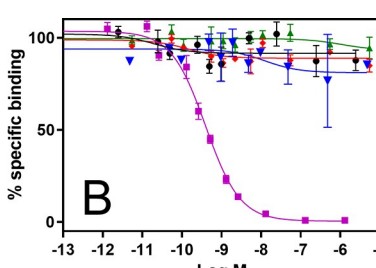 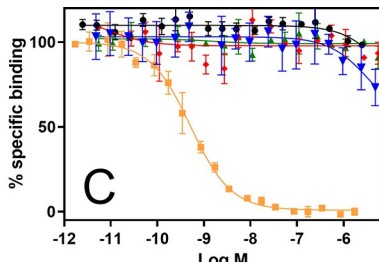

**Fig 3.** Binding curves for (**A**) IGF2R, (**B**) IGF1R, (**C**) IR-A. Inhibition of binding of human [$^{125}$I]-monoiodotyrosyl-ligand to corresponding receptor by IGF2 (cyan), IGF1 (magenta), insulin (orange), peptide **1** (black), peptide **2** (red), peptide **3** (blue), peptide **4** (green). Representative binding curve for each hormone or analog is shown, n ≥ 3 independent experiments with 2 replicates.

induced mitogenesis of osteoblasts involves activation of p42/44 Erk proteins (Erk 1/2) [9]. Their findings indicated that preptin can induce significant mitogenesis in osteoblasts via Erk 1/2 activation. They demonstrated this effect in primary cultures of rat and human osteoblasts, as well as in MC3T3-E1 cells and the human osteoblast-like cell line SaSO-2 [9]. Thus, in addition to determining Erk 1/2 activation, we also focused on the Akt and PI3K pathways to gain further insights into the biological properties of preptins and their analogs (Fig 4 and S44 Fig in S1 Data).

We studied phosphorylation of Akt, Erk 1/2 and PI3K 110α proteins by preptins or their analogs on R- fibroblasts (Fig 4A), preosteoblasts (Fig 4B), osteoblasts (Fig 4C) and MIN6 cells (Fig 4D), i.e. cells that should be sensitive to preptin. The only significant changes we observed in our experiments were in phosphorylation of Akt. All our stapled preptin analogs (peptides **6**–**11**) that we prepared induced significantly lower Akt phosphorylation on preosteoblasts (MCT3T3-E1 cells) compared to vehicle (Fig 4B). Significantly lower phosphorylation of Akt on MCT3T3-E1 cells was also observed for the C-terminal preptin analogs (peptides **12** and **13**). All other results show no significant changes in the ability to trigger any of the studied signaling pathways in any type of cell lines used in this study. This means that neither preptin nor its analogs seem to be able to induce any significant response in cells that contain either the IGF2 receptor or bone or pancreatic beta cells. On the other hand, stapled and shortened fragments **6**–**13** seem to have some apparent inhibitory effects (at $10^{-8}$ M) on Akt signaling in MC3T3-E1 preosteoblasts. However, the molecular mechanism of the effect is unknown.

## High concentrations of preptin and its analogs can stimulate a change in calcium levels

It is well known that $Ca^{2+}$ regulates numerous physiological cellular processes as a second messenger. It plays a vital role in the regulation of cell proliferation, and its increased level is critical for secretagogue-induced insulin release in pancreatic beta cells [39]. Since preptin has been suggested as a phase 2 insulin secretion influencer [7], measuring changes in intracellular calcium was another method we used to study the effects of preptin on different cell types (Fig 5). If preptin or its analogs were proficient insulin secretagogues or had any effect on bone cells, we would expect changes in intracellular calcium levels in these cells. We observed in our previous paper that native preptin and some of its stapled fragments were able to induce changes in intracellular calcium levels in U-2 OS cells [21]. Here we used a different, more sensitive method, by the use of four different cell lines (Fig 5). We observed significant changes in calcium levels only in bone U-2 OS cells, which we used in our previous study [21]. The significant increase in intracellular calcium levels was detected for native rat preptin (peptide **19**), mouse preptin with a conformationally rigid Aib amino acid at the position 20 (where proteolytic cleavage of the peptide likely occurs) (peptide **15**), and a fragment comprising amino acids 13–27 of mouse preptin with an amidated C-end (peptide **18**). On the contrary, almost no effect on intracellular calcium levels was observed for peptides **6**–**11**, which are stapled peptides. Generally, our results agree with previous publications, where forcing the secondary structure always reduced the biological activity [10].

The only significant effect on calcium levels in MIN6 cells was observed for peptide **17**, which completely inhibited calcium secretion. This result is interesting, but it is not consistent with results in other cell lines, nor is it consistent with the effect of a similar preptin fragment (peptide **16**) and we are unable to interpret this result. We did not observe any significant difference between DMSO-treated cells (negative control) and our peptides in R- or MC3T3-E1 treated cells. Our results again confirmed that the only sensitive cell line in which we could observe changes in intracellular calcium levels were U-2 OS cells and all were non-significant.

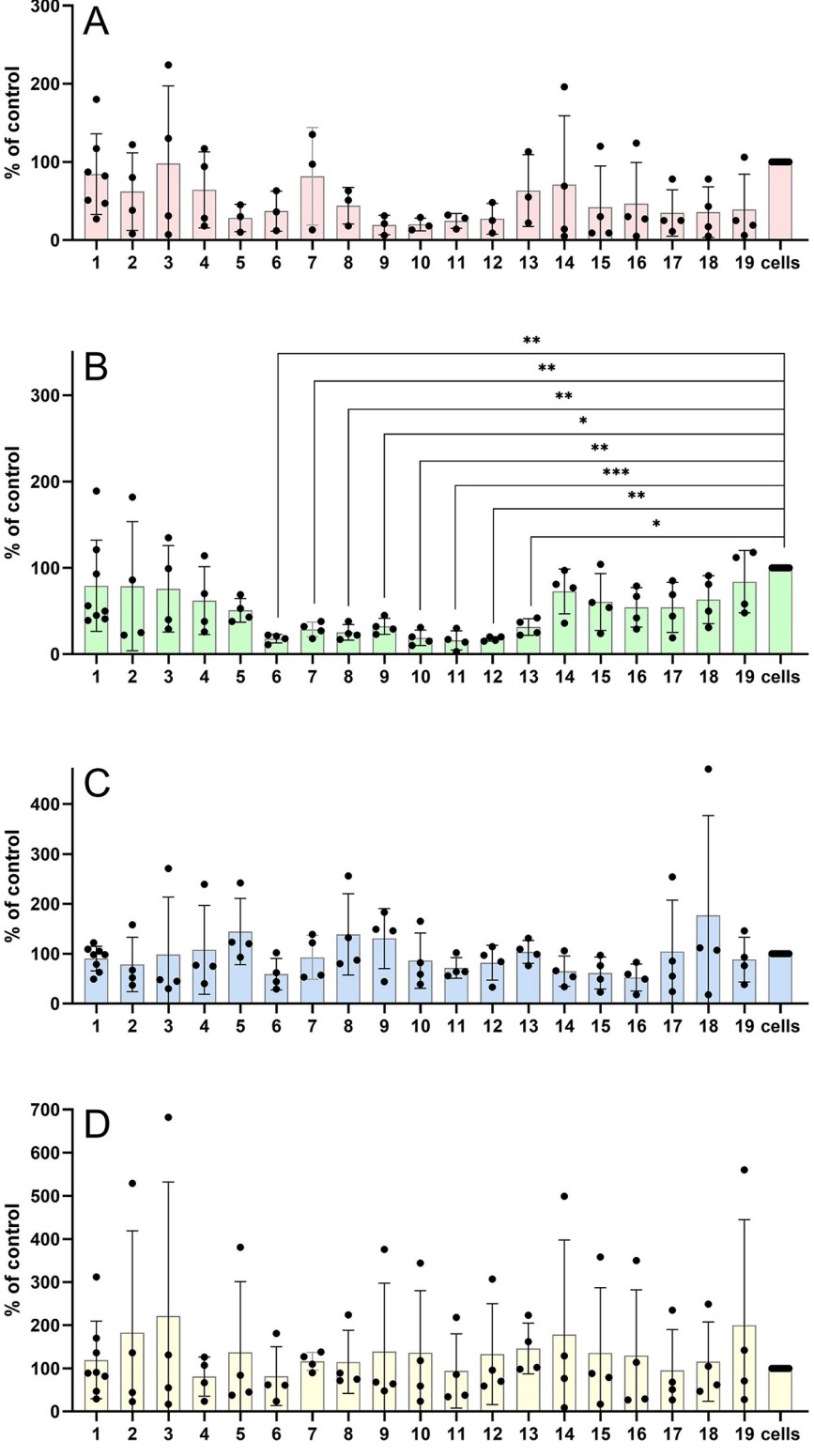

**Fig 4. Stimulation of phosphorylation of Akt by preptins (at 10⁻⁸M).** Stimulation of phosphorylation of intracellular Akt proteins in murine fibroblasts with deleted *Igf1r* gene (R- cells) (**A**), in MC3T3-E1 preosteoblasts (**B**), in U-2 OS osteoblasts (**C**) and MIN6 pancreatic beta cells (**D**) by respective preptins or preptin analogs. Data are presented as means ± S.D., relative to the signal in non-stimulated cells. Significant differences between marked values determined using Ordinary one-way ANOVA (Dunnett´s multiple comparisons test), *p < 0.05, **p < 0.01, ***p < 0.001, ****p < 0.0001.

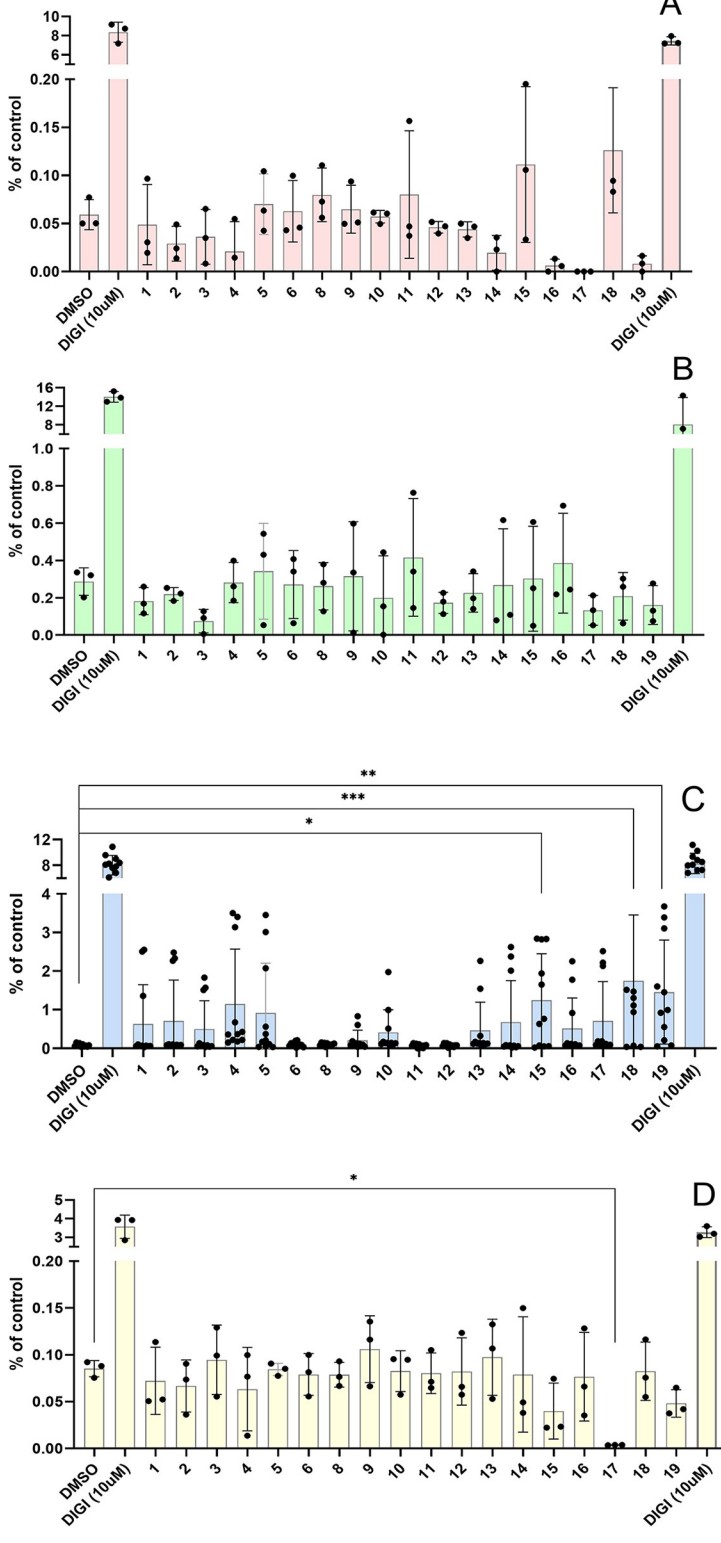

**Fig 5. Changes in intracellular calcium after stimulation with preptin and its analogs at the concentration $10^{-4}$ M.** Murine fibroblasts with deleted *igf1r* gene (R- cells) (**A**), MC3T3-E1 preosteoblasts (**B**), U-2 OS osteoblasts (**C**) and MIN6 pancreatic beta cells (**D**). Data are presented as means ± S.D., relative to the signal in non-stimulated cells. Significant differences between marked values determined, using Ordinary one-way ANOVA (Dunnett´s multiple comparisons test), *p < 0.05, **p < 0.01, ***p < 0.001, ****p < 0.0001.

## Preptin and its analogs/fragments do not affect cAMP levels in different cell lines

Changes in cAMP levels is another parameter that could reflect the effects of preptin or its analogs. The second messenger cAMP is one of the most important cellular signaling molecules, with central function including regulation of insulin secretion from the pancreatic β-cells. It is generally considered as an amplifier of insulin secretion, triggered by elevation of intracellular calcium in the β-cells [40]. Cyclic nucleotides such as cAMP are also known to have a dramatic effect on bone cells and the cAMP pathway, as well as the $Ca^{2+}$-PKC pathway, and influence osteoblast proliferation and differentiation [41]. For this reason, we measured changes in cAMP levels to complement data on changes in $Ca^{2+}$ levels and to provide additional information on the possible effects of preptin or its analogs (Fig 6). We used the cAMP activator forskolin as a positive control. We measured changes in cAMP levels in three different cell lines, cAMP-HEK cells (Fig 6A), U-2 OS osteoblasts (Fig 6B) and pancreatic MIN6 cells (Fig 6C). Unfortunately, the positive control many times exceeds the changes that we observed after stimulation with our peptides and, similarly to the measurement of changes in intracellular calcium levels, we did not see any significant changes. The only significant decrease of cAMP was observed after the treatment with peptide **13** in U-2 OS osteoblasts. Neither native preptin nor analogs showed any significant effect on this signaling pathway.

## Radioimmunology assay (RIA) detect no changes in insulin secretion from MIN6 pancreatic beta cells

Using a radioimmunoassay, we wanted to confirm the results of previous studies that suggested an effect of preptin or its analogs on insulin secretion in pancreatic beta cells [7,22]. The literature suggests that preptin stimulates the second phase of insulin secretion [7,22]. Cheng et al. [22] observed an increase in insulin secretion at a glucose concentration of 25 mmol/l and Buchanan et al. [7,8] used 10 mmol/l. Moreover, Buchanan et al. [8] demonstrated that only rat preptin was able to increase insulin secretion on the permanent mouse β-cell line βTC6-F7, whereas human preptin was not. We used a permanent pancreatic beta cell line MIN6 that shows sufficient insulin production and responsiveness [33,42]. We used mouse preptin and a basal glucose concentration of 10 mmol/l for this experiment. Nevertheless, we were unable to confirm the hypothesis that preptin can activate insulin secretion from pancreatic beta cells, and, even after repeated measurements, we did not observe a significant increase in insulin secretion despite using very high concentrations of preptin (Fig 7). Although our model MIN6 cells accumulate sufficient insulin secretory granules, they may be less sensitive than primary cultures. Buchannan et al. demonstrated a positive effect of preptin on insulin secretion, both in the cell line and in the whole isolated rat pancreas. However, in our experiments, although we could see some minor trends, we did not observe significant results, similarly to the signaling experiments above.

## Effect of preptin on osteoblast differentiation in hBMSCs

As preptin was previously shown to stimulate osteoblast differentiation [10], we tested the effect of our preptin analogs on osteoblast differentiation of hBMSC-TERT by measurement of alkaline phosphatase (ALP) activity (a key enzyme that is important for osteoblast differentiation) and Alizarin S staining (AZR) to visualize calcified matrix (Figs 8 and 9). Although hBMSCs stimulated with IGF2 and preptin showed signs of higher ALP activity compared to unstimulated cells, the only significant changes were observed when cells were stimulated with 100 ng/ml and 1000 ng/ml IGF1. Similar results were obtained in the study by Bosetti et al.

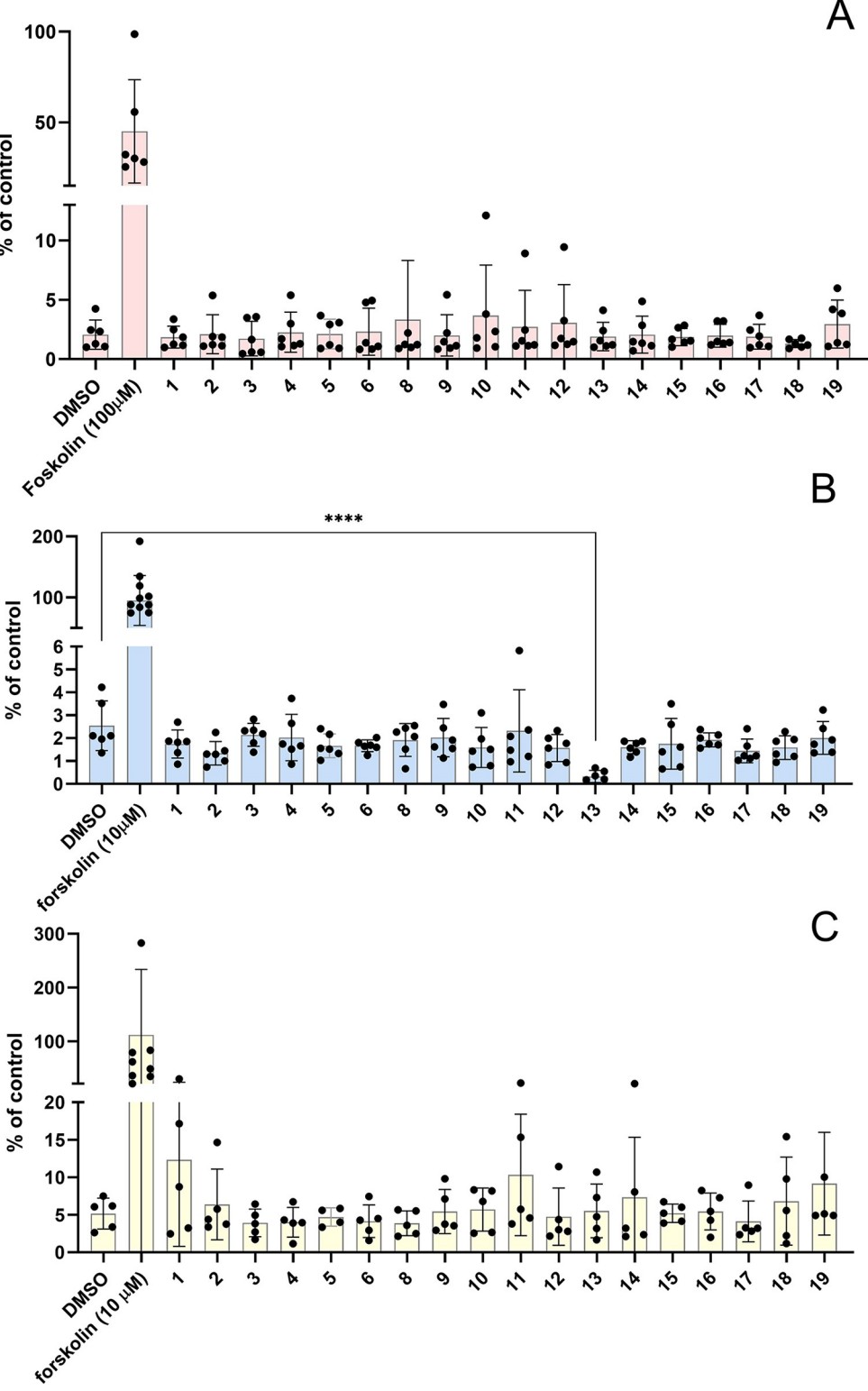

**Fig 6. Changes in cAMP levels after stimulation with preptin and its analogs at a concentration of $10^{-4}$ M.** cAMP-HEK cells (**A**), U-2 OS osteoblasts (**B**) and MIN6 pancreatic beta cells (**C**). Data are presented as means ± S.D., relative to the signal in non-stimulated cells. Significant differences between marked values, determined using Ordinary one-way ANOVA (Dunnett´s multiple comparisons test), ****$p < 0.0001$.

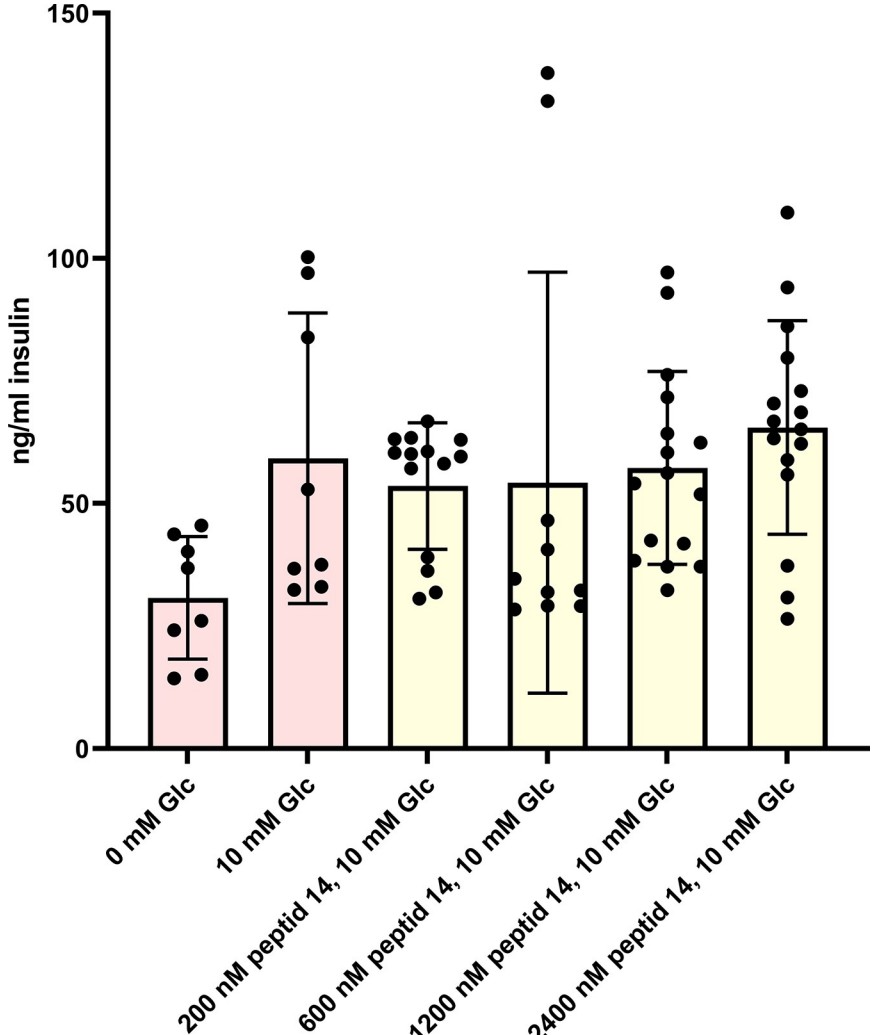

**Fig 7. Stimulation of insulin secretion by various concentrations of native mouse preptin (peptide 14) on pancreatic MIN6 cells.**

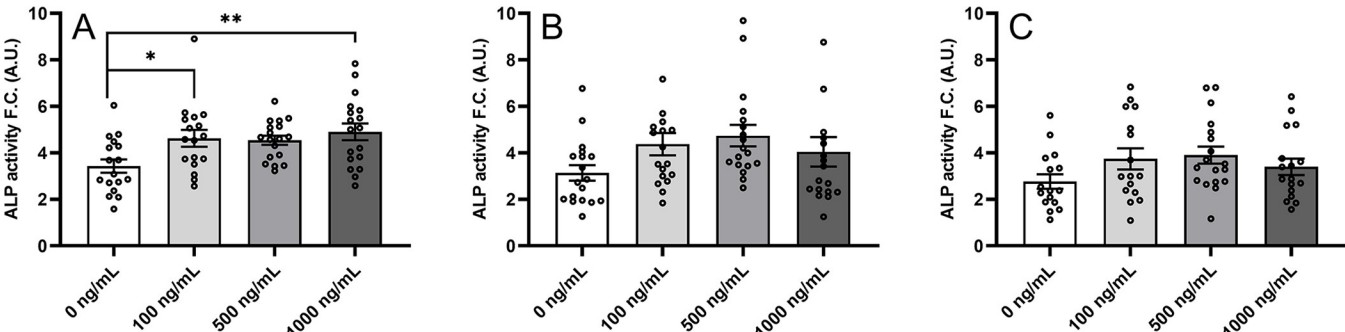

**Fig 8.** Differentiation of hBMSC-TERT osteoblasts after treatment with IGF1 (A), IGF2 (B), and preptin (C). The data were evaluated using quantification of alkaline phosphatase (ALP) activity, normalized to cell viability presented as a fold change (F.C.) over non-induced cells (day 7), n = 3 independent experiments with 6 replicates. Data are presented as mean ± SEM, Ordinary one-way ANOVA (Dunnett´s multiple comparisons test), *p < 0.05, **p < 0.01. A. U. means arbitrary units.

[17], where the only significant change in ALP activity in human primary osteoblasts after a similar time period (8 days versus 7 days in our case) was observed with IGF1, but IGF2 and preptin showed no significant effects.

Furthermore, our results with Alizarin staining also show the largest changes after treatment with IGF1 (Fig 9A) as compared to effects of IGF2 and preptin (Fig 9B and 9C).

## Conclusions

In our study, we aimed to investigate the role of preptin, a 34-amino acid peptide derived from pro-IGF2 on cellular physiology. We synthesized 16 different forms of preptin, ranging from native human, rat and mouse preptins to analogs or fragments. These compounds were tested in a variety of assays and results compared to previously published data on preptins. Our first goal was to investigate if preptin is able to interact with any of the receptors to which IGF2 can bind. Contrary to previous indications [22], our experiments did not reveal any significant binding of preptin to IGF2-sensitive receptors, including IGF2R, IR-A, and IGF1-R. Similarly, radiolabeled preptin failed to exhibit relevant binding in target tissues previously associated with preptin activity. Despite these findings, we continued our investigation to explore the effects of preptin and its analogs on intracellular processes.

Several studies reported a positive growth effect of preptin on osteoblasts [9,43,44] and some did not [10,17], even using very similar cellular systems. Since the results from other studies were not entirely conclusive, we investigated what intracellular processes might be involved. In general, the positive effect on bone cells is commonly accompanied by an increase in Erk 1/2 signaling, an increase in ALP, and an effect on intracellular calcium or cAMP levels.

To study intracellular signaling, we used cell lines that should be sensitive to preptin. However, the only significant changes we observed were a decrease in Akt protein phosphorylation in MC3T3-E1 preosteoblasts after treatment with our stapled preptins. Despite several promising results published in the previous study [9], changes in phosphorylation of Erk 1/2 or PI3K p110a protein were insignificant in all cell types we probed and for all our preptins. In measuring changes in intracellular calcium ion levels, the only significant changes we observed in U-2 OS cells were also used in our previous study [21]. As in our previous study, we found that stapled peptides have an impaired ability to affect intracellular calcium levels. On the other hand, two full-length preptins and one middle fragment significantly increased intracellular calcium levels. Changes in cAMP were not significant over all analogs and all cells, apart from a significant decrease in peptide **13** in U-2 OS cells, which can probably be interpreted as an experimental artefact. Moreover, for $Ca^{2+}$ and cAMP measurements, we had to use high non-physiological concentrations of preptins to detect any effects. All the experiments were firstly done in 10–100 nM concentrations. However, later we proceeded to apply compounds in millimolar concentrations.

Preptin was originally found in pancreatic beta cells and was thought to be an insulin secretagogue. Unfortunately, our results with glucose-stimulated insulin secretion also failed to show a significant increase in insulin secretion, even in the presence of 10 mM glucose. Despite repeated attempts, we were unable to confirm the hypothesis with preptin as a secretagogue [7,8,22]. Moreover, results with hBMSCs showed that neither preptin nor IGF2 significantly increased the osteoblast differentiation, with the only significant changes observed for IGF1.

Preptin has been the subject of a relatively large number of published studies evaluating its higher or lower levels in the context of different diseases or in attempting to observe differences between normal and preptin KO mice [45,46]. The results are very inconclusive. One of the fundamental aspects is that almost no publications consider the fact that higher levels of preptin correlate inherently with higher levels of IGF2 or IGF2 proforms. Since preptin is

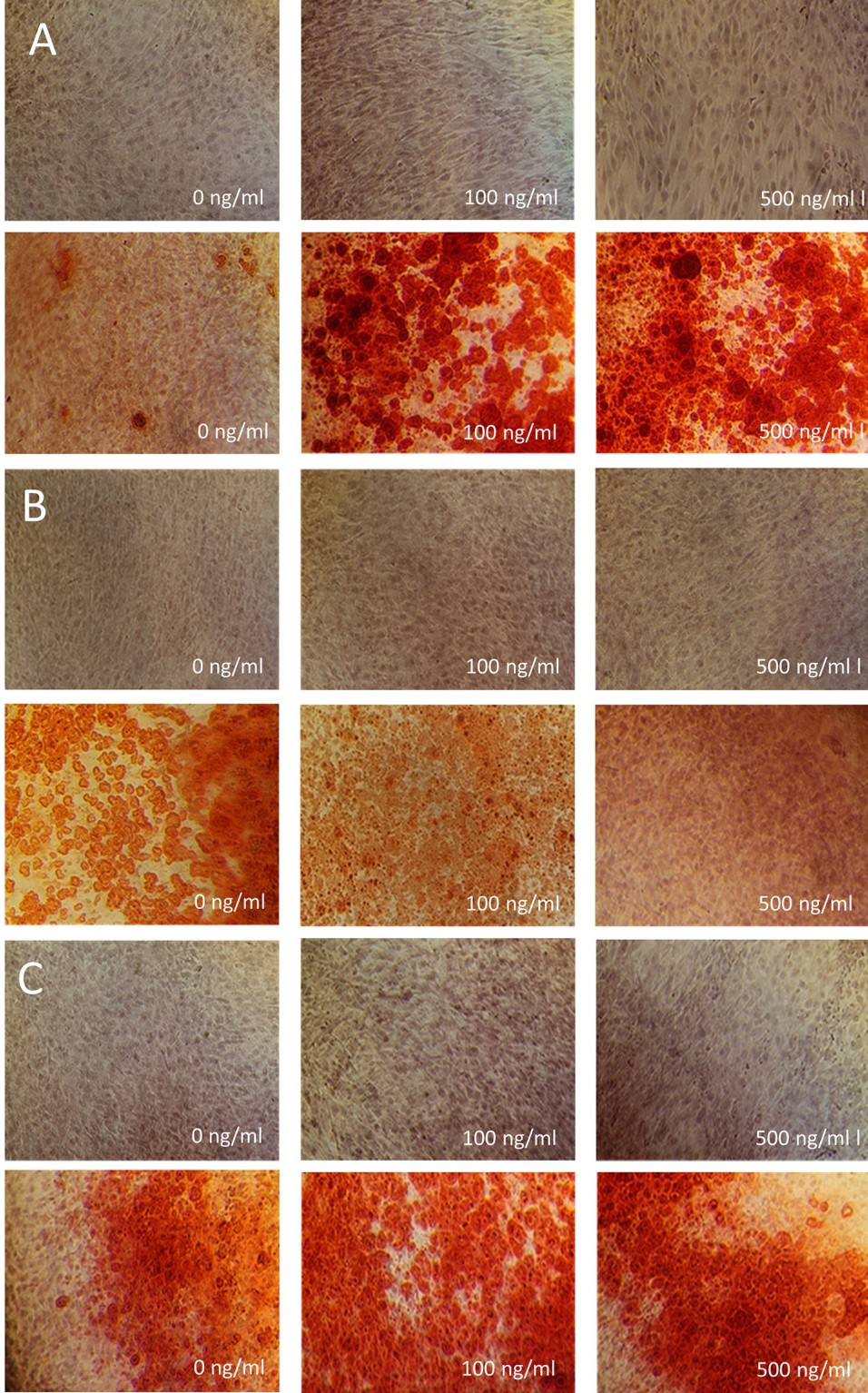

**Fig 9. Staining of hBMSC-TERT differentiated into osteoblasts with alizarin S.** Staining of hBMSC-TERT differentiated into osteoblasts for 10 days *in vitro* with alizarin S. Cells were treated with increasing concentrations of IGF1 (A), IGF2 (B) and rat preptin 19 (C). The upper panels in all cases show undifferentiated osteoblasts treated with the appropriate concentration of the compound; the lower panels always show differentiated osteoblasts treated with the adjacent concentration of the compound.

separated from pro-IGF2, its higher levels will also mean higher levels of IGF2 in the body. Therefore, this can mean that some of the symptoms contributed to actions of preptin may be due to higher levels of IGF2 that were not tracked in almost any study. IGF2 or various pro-forms of IGF2 are generally known to be associated with the occurrence of various diseases. In this regard, we would like to highlight the study of Buckels et al. [45,46], where different ELISA kits for preptin, their accuracy and sensitivity, were addressed. It appears from the study that these ELISA kits may also measure, for example, big-IGF2(104) or pro-IGF2(156), since preptin is a part of these proteins.

In conclusion, since no significant phenotype changes were observed in preptin KO mice [45,46], it is very likely that preptin is not a relevant biologically active molecule and the observed effects are probably due to high concentrations of preptin in the assays or to the effects of IGF2 or IGF2 proforms present in samples or experimental models. Our data rather support this hypothesis.

## Supporting information

**S1 Data. Detailed description of the synthesis of precursor amino acids for click and olefin metathesis reactions (Schemes S1, S2, S1-S4 Figs), analytical data for peptides 1–19 (S5-S42 Figs, S1-S5 Tables), binding curves of iodinated peptide 14 to cells (S43 Fig) and data showing stimulation of PI3K p110α in cells by preptins 1–19 (S44 Fig).**
(DOCX)

## Author Contributions

**Conceptualization:** Lucie Mrázková, Jiří Jiráček, Lenka Žáková.

**Data curation:** Lucie Mrázková, Jakub Kaminský, Lenka Žáková.

**Formal analysis:** Lucie Mrázková, Michaela Tencerová, Jakub Kaminský, Lenka Žáková.

**Funding acquisition:** Michaela Tencerová, Jiří Jiráček, Lenka Žáková.

**Investigation:** Lucie Mrázková, Marta Lubos, Jan Voldřich, Erika Kužmová, Denisa Zrubecká, Petra Gwozdiaková, Miloš Buděšínský, Seiya Asai, Aleš Marek, Jan Pícha, Michaela Ferenčáková, Glenda Alquicer Barrera, Jakub Kaminský, Lenka Žáková.

**Methodology:** Lucie Mrázková, Marta Lubos, Jan Voldřich, Erika Kužmová, Michaela Tencerová, Jakub Kaminský, Jiří Jiráček, Lenka Žáková.

**Project administration:** Michaela Tencerová, Jiří Jiráček, Lenka Žáková.

**Resources:** Lenka Žáková.

**Software:** Lenka Žáková.

**Supervision:** Michaela Tencerová, Jiří Jiráček, Lenka Žáková.

**Validation:** Lenka Žáková.

**Visualization:** Lenka Žáková.

**Writing – original draft:** Lucie Mrázková, Marta Lubos, Michaela Tencerová, Michaela Ferenčáková, Jakub Kaminský, Jiří Jiráček, Lenka Žáková.

**Writing – review & editing:** Lucie Mrázková, Erika Kužmová, Michaela Tencerová, Michaela Ferenčáková, Jakub Kaminský, Lenka Žáková.

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
