## [Decision Letter · Decision Letter 0]

21 Jun 2024

PONE-D-24-18330THE FINAL WALK WITH PREPTIN?PLOS ONE

Dear Dr. Žáková,

Thank you for submitting your manuscript to PLOS ONE. After careful consideration, we feel that it has merit but does not fully meet PLOS ONE’s publication criteria as it currently stands. Therefore, we invite you to submit a revised version of the manuscript that addresses the points raised during the review process.

We look forward to receiving your revised manuscript.

Kind regards,

Haim Werner

Academic Editor

PLOS ONE

“The work was supported by the Czech Science Foundation (grant No. 19-14069S (to LZ) and 22-12243S (to MT)), by the National Institute for Research of Metabolic and Cardiovascular Diseases (Program EXCELES, ID Project No. LX22NPO5104) - Funded by the European Union – Next Generation EU and by the Academy of Sciences of the Czech Republic (Research Project RVO:6138963, support to the Institute of Organic Chemistry and Biochemistry).”

“The work was supported by the Czech Science Foundation (grant No. 19-14069S (to LZ) and 22-12243S (to MT)), by the National Institute for Research of Metabolic and Cardiovascular Diseases (Program EXCELES, ID Project No. LX22NPO5104) - Funded by the European Union – Next Generation EU and by the Academy of Sciences of the Czech Republic (Research Project RVO:6138963, support to the Institute of Organic Chemistry and Biochemistry).”

“The work was supported by the Czech Science Foundation (grant No. 19-14069S (to LZ) and 22-12243S (to MT)), by the National Institute for Research of Metabolic and Cardiovascular Diseases (Program EXCELES, ID Project No. LX22NPO5104) - Funded by the European Union – Next Generation EU and by the Academy of Sciences of the Czech Republic (Research Project RVO:6138963, support to the Institute of Organic Chemistry and Biochemistry).”

Reviewers' comments:

Reviewer's Responses to Questions

**Comments to the Author**

1. Is the manuscript technically sound, and do the data support the conclusions?

Reviewer #1: Yes

Reviewer #2: Yes

2. Has the statistical analysis been performed appropriately and rigorously? 

Reviewer #1: Yes

Reviewer #2: Yes

3. Have the authors made all data underlying the findings in their manuscript fully available?

Reviewer #1: Yes

Reviewer #2: No

4. Is the manuscript presented in an intelligible fashion and written in standard English?

Reviewer #1: Yes

Reviewer #2: Yes

5. Review Comments to the Author

Reviewer #1: Preptin previously was thought to have specific biological activities, such as stimulating bone growth, but the authors have done a thorough analysis and revealed no significant biological activity associated with preptin or its

analogs, suggesting the associated functions of preptin may be influenced by factors, such as higher

levels of IGF2 or IGF2 prohormones present in tissues. The authors claim has taken in the context of the previous literature, which when analysing their interesting and thorough data, I support their assumptions. The data is clearly reported and the manuscript written very clearly.

The authors synthesized 16 different forms of preptin, including human and rodent analogues or fragments and they assayed them to test the interaction with the receptors to which IGF2 can bind and cells from target tissues

previously associated with preptin activity, such as osteoblasts. Contrary to previous published indications no significant binding or cellular effects of preptin were seen. The explanation that as preptin is separated from pro-IGF2, then higher levels of preptin will also mean higher levels of IGF2 so that some of the symptoms contributed to actions of preptin may be due to higher levels of IGF2 is certainly feasible. This is backed up by the lack of phenotypic changes that were observed in preptin KO mice.

Reviewer #2: This manuscript describes the chemical synthesis of preptin and a series of preptin analogues and analysis of their in vitro biological activities. There is some controversy in the literature as to the biological action of preptin, which arises during proteolytic maturation of the pro-IGF2 peptide. This study sought to clarify some of these controversies. The peptides were successfully chemically synthesised. Subsequent in vitro analyses, which were appropriate for gaining an understanding of action via the IGF/insulin receptors and biological processes downstream, essentially failed to demonstrate any significant activities for all of the peptides. The conclusions are sound and it is commendable for the authors to submit this for publication such that some further clarity is provided to the field.

Comments/suggestions:

The last two sentences of the introduction are vague and could be written more concisely.

Page 3 please clarify “its levels negatively correlate with bone mineral density and osteoporosis” – correlate with lower or higher bone mineral density??

Page 3 please clarify “to approach?? the receptor for preptin, we performed several binding experiments”

Page 10 last paragraph “after overnight starvation in a clean culture medium without serum” – not “cultivated”

Page 14 first paragraph of results. “confirmed” – not “conformed”

It is noteworthy that some of the peptide treatments result in a lower Akt phosphorylation than the control (Fig 3b). What is added to the control – is it the same vehicle as used for the peptides? Does this mean there is something inhibitory in the vehicle control?

What was the rationale for using 10-8M peptide and how does this relate to the binding affinities derived in Fig 2?

General comments on immunoblots: Was a positive control used to demonstrate the ability to activate Akt, Erk and PI3K? Were responses normalised to demonstrate that no changes in expression of Akt, Erk and PI3K have occurred during stimulation?

No blots are provided and at minimum representative blots should be included (perhaps in the supplementary section)

In the insulin secretion assay, it does not appear that the cells are producing significant amounts of insulin in response to the positive control (10mM glucose). Is this the case and if so how is it possible to comment then on the activity of the preptin peptides?

Page 20 last sentence “(REF)”

Figure 8 legend the English is not clear “treated with the adjacent? concentration of the compound”

I am not sure if conclusions can be drawn from the alizarin S staining as there is considerable variability at the 0 ng/ml concentration treatments. It is hard to see how any statistical differences could be derived with this variability.

6. PLOS authors have the option to publish the peer review history of their article (what does this mean?). If published, this will include your full peer review and any attached files.

Reviewer #1: **Yes: **Jillian Cornish

Reviewer #2: **Yes: **Briony Forbes

---

## [Author Response · Author response to Decision Letter 0]

2 Aug 2024

We would like to thank you and the reviewers for the careful assessment of our work and detailed comments. These comments were very valuable, helping us to clarify the major points of the paper and to remove ambiguities. We hope that we have addressed all concerns raised by the reviewers and implemented their suggestions effectively. Our point-by-point responses to the reviewers’ comments are as follows. All changes, which we made in the original manuscript are highlighted with track changes. 

Reviewers' comments:

Reviewer #1:

We thank Jillian Cornish, as one of the first authors working on preptin, for her positive review of our manuscript. We found no specific comments in her review that needed to be responded to.

Reviewer #2:

Thanks to Briony Forbes for her excellent and detailed review of our manuscript, and here are the answers to her questions:

The last two sentences of the introduction are vague and could be written more concisely.

We've rewritten it.

Page 3 please clarify “its levels negatively correlate with bone mineral density and osteoporosis” – correlate with lower or higher bone mineral density??

 We have edited the text into a more readable version.

Page 3 please clarify “to approach?? the receptor for preptin, we performed several binding experiments”

 We've corrected it.

Page 10 last paragraph “after overnight starvation in a clean culture medium without serum” – not “cultivated”

 It has been corrected.

Page 14 first paragraph of results. “confirmed” – not “conformed”

 It has been corrected.

It is noteworthy that some of the peptide treatments result in a lower Akt phosphorylation than the control (Fig 3b). What is added to the control – is it the same vehicle as used for the peptides?

 The reviewer is right here, sometimes the phosphorylation of Akt by several of our peptides does not seem to reach control values. This is due to the fact that the signal was very weak and there was a relatively large range of values for these weak signals, so the average values can fall below the control values. 

The vehicle was the same for all peptides and controls. 

The only values that seem inhibitory are for peptides 6-13, which are very short or stapled analogues of preptin, which are really significantly lower, and we discuss their low effect in the Discussion section.

What was the rationale for using 10-8M peptide and how does this relate to the binding affinities derived in Fig 2?

The 10-8M concentration was used for three reasons. First, to get closer to physiological conditions. Besides, we did signaling at higher concentrations, but the results were the same. As the reviewer correctly points out the second reason for using the 10 nM concentration results from the usual Kd values of insulin-like peptides (insulin and IGFs), which are not very different from 10 nM The third reason was the sensitivity of signal detection in Western blots, which is not sufficient for efficient detection of hormone activities below 10 nM. 

Was a positive control used to demonstrate the ability to activate Akt, Erk and PI3K? Were responses normalised to demonstrate that no changes in expression of Akt, Erk and PI3K have occurred during stimulation?

 In our first experiments, we used insulin and IGFs as a positive control and they signaled highly. Compared to these controls, our signals were very weak and when we repeated the experiments already with respective analogues, we did not use these positive controls and only looked at any effect compared to control unstimulated cells.

No blots are provided and at minimum representative blots should be included (perhaps in the supplementary section)

Representative blots were added (Figure S46) to the supplemental material. 

In the insulin secretion assay, it does not appear that the cells are producing significant amounts of insulin in response to the positive control (10mM glucose). Is this the case and if so how is it possible to comment then on the activity of the preptin peptides?

It is true that 10 mM glucose does not come out significantly in our reported RIA essay, although the graph shows an increase. We do the RIA test with and with other secretagogues quite routinely and normally insulin secretion is significantly increased after stimulation with 10 mM glucose. Here, the non-significant behaviour is given by the ANOVA statistical test we used. In this test, the results are influenced by the whole evaluation group and not just two columns. If we used a t-test between only 0 mM and 10 mM glucose, the result would be significant (P = 0.0253 (i.e. P < 0.05)). 

Page 20 last sentence “(REF)”

 It has been corrected.

Figure 8 legend the English is not clear “treated with the adjacent? concentration of the compound”

 It has been corrected.

I am not sure if conclusions can be drawn from the alizarin S staining as there is considerable variability at the 0 ng/ml concentration treatments. It is hard to see how any statistical differences could be derived with this variability.

 Yes, we agree that Alizarin staining is difficult to evaluate statistically. In our case, Alizarin staining serves as a confirmation of ALP measurement, which, like Alizarin staining, shows the highest effect for IGF1, an effect also known from the literature. Neither IGF2 nor preptin achieve this effect and statistical increase in ALP.

The paragraph related to funding has been removed from the manuscript and information on funding has been added to the Funding Statement section of the online submission form.

We hope that the response together with changes and corrections will fulfill the reviewer’ as well as your requirements, and that this revised manuscript could be considered for publication in the PloS One. Please do not hesitate to contact me in case of any further queries.

---

## [Editor Report · Decision Letter 1]

19 Aug 2024

THE FINAL WALK WITH PREPTIN?

PONE-D-24-18330R1

Dear Dr. Žáková,

We’re pleased to inform you that your manuscript has been judged scientifically suitable for publication and will be formally accepted for publication once it meets all outstanding technical requirements.

Kind regards,

Haim Werner

Academic Editor

PLOS ONE

Additional Editor Comments (optional):

Authors have satisfactorily addressed reviewer's comments.
---

## [Editor Report · Acceptance letter]

4 Sep 2024

PONE-D-24-18330R1 

PLOS ONE

Dear Dr. Žáková, 

I'm pleased to inform you that your manuscript has been deemed suitable for publication in PLOS ONE. Congratulations! Your manuscript is now being handed over to our production team.

Kind regards, 

on behalf of

Dr. Haim Werner 

Academic Editor

PLOS ONE